# Effects of Maturation on Lower Limb Neuromuscular Asymmetries in Elite Youth Tennis Players

**DOI:** 10.3390/sports7050106

**Published:** 2019-05-08

**Authors:** Marc Madruga-Parera, Daniel Romero-Rodríguez, Chris Bishop, Maria Reyes Beltran-Valls, Alex T. Latinjak, Marco Beato, Azahara Fort-Vanmeerhaeghe

**Affiliations:** 1EUSES Health and Sport Sciences School, University of Girona, 17190 Girona, Spain; danirrphysco@gmail.com (D.R.-R.); a.latinjak@uos.ac.uk (A.T.L.); 2EUSES Health and Sport Sciences School, University Rovira Virgili, 43870 Amposta, Spain; 3Faculty of Science and Technology, London Sport Institute, Middlesex University, London NW4 BT, UK; C.Bishop@mdx.ac.uk; 4LIFE Research Group, Department of Education, University Jaume I, 12071 Castellón de la Plana, Spain; reyes.bel@outlook.com; 5School of Science, Technology and Engineering, University of Suffolk, Ipswich IP3 0AT, UK; m.beato@uos.ac.uk; 6Department of Sports Sciences, Ramon Llull University, FPCEE and FCS Blanquerna, 08025 Barcelona, Spain; afortvan@gmail.com

**Keywords:** inter-limb differences, vertical jump, balance, change of direction

## Abstract

Neuromuscular asymmetries have been previously associated with reduced performance. Similarly, maturation has shown that youth athletes may experience a loss of motor control, which could also lead to compromised physical performance. The present study aimed to evaluate and quantify the level of asymmetry among chronological and maturational groups. Forty-one youth tennis players performed the single leg countermovement jump (SLCMJ), star excursion balance test (SEBT) and a change of direction speed (CODS) test. Differences were found between the strongest and weakest limbs across all tests (*p* < 0.001), and also for SEBT in the posteromedial direction (*p* = 0.02), SEBT composite score (*p* < 0.01) in maturation groups, and for SEBT posterolateral direction (*p* = 0.03) and SEBT composite score (*p* = 0.01) in chronological groups. The SLCMJ showed the largest inter-limb asymmetries for the circa peak height velocity (PHV) group (19.31 ± 12.19%) and under-14 (U14) group (17.55 ± 9.90%). Chronological and maturation groups followed similar trends for inter-limb asymmetries, but the biological index showed larger asymmetry scores in the jumping test at PHV compared to that found in the chronological group (U14). These results show that inter-limb differences may be heightened during PHV. Practitioners can use this information to inform the decision-making process when prescribing training interventions in youth tennis players.

## 1. Introduction

Tennis is an intermittent sport characterized by repetitive high-intensity efforts (e.g., accelerations, decelerations) during competition with an average duration of 90 min per match [1]. Thus, the development of multiple physical qualities [2] is essential for a tennis players’ athletic development. More specifically, acceleration and braking skills are key characteristics in tennis performance and require a combination of musculoskeletal, neural, and coordinative components simultaneously [3]. It should also be noted that tennis players perform more actions on their forehand side [4], and due to the repetitive sport-specific demands, this represents an intrinsic risk factor for potential injuries [5]. Therefore, the development of lower limb asymmetries should be expected and taken into consideration in any injury prevention strategy in tennis.

Recent evidence has shown that larger asymmetries are a key factor to consider in the return to play process after injury [6], are linked to reduced neuromuscular control during hopping tasks [7], and have a detrimental effect on measures of physical performance [8,9]. Recently, Bishop et al. [8] showed that greater lower-limb asymmetries in jump height were associated with reduced 5, 10, 20 m sprint and vertical jump performance in youth female soccer athletes. Furthermore, Sannicandro et al. [9] showed that young soccer players (mean age = 11.2 years) displayed inter-limb asymmetries in the lateral hop test (18.5%), which was notably larger than that of the single or triple hop tests, and greater (*p* < 0.05) than that of younger players (mean age = 9.1 years). Thus, since asymmetries have been shown to be associated with reduced physical performance and potentially increase as youth athletes approach maturation, further research is warranted on the interaction between asymmetry and maturation in youth athletic populations.

In tennis, movements, such as change of direction speed (CODS) are frequently required [4], and given the unlikely nature of this high-intensity action occurring an equal number of times on each limb, will likely contribute to the development of neuromuscular asymmetries over time [10]. To the authors’ knowledge, only one previous study has investigated the presence of lower-limb neuromuscular asymmetries in strength and speed tasks in youth tennis athletes [11]. These authors used a six-week training intervention consisting of lower limb balance and strength-based exercises performed twice a week. Results showed a reduction in asymmetry during the single hop (9.0 to 3.7%; *p* < 0.001), lateral hop (10.8 to 3.2%; *p* < 0.001), and CODS (7.2 to 2.7%; *p* < 0.05) tests. While useful, this study did not take into consideration the maturation levels of players; thus, it is possible that such analysis would have resulted in altered between-limb differences, given the changes in motor control during these stages of development [12].

Differences in the morphological profiles of young tennis players and how these values can affect coordinative abilities have been identified among chronological age categories [13], especially during adolescence, which is one of the most vulnerable times when youth athletes may be subject to injury [14]. Van der Sluis et al. [14] recorded 178 injuries during three years among 26 youth soccer players (82% in the lower limbs), being higher in the peak height velocity (PHV) group in comparison to the pre-PHV and post-PHV groups. This can be explained by the negative effects on neuromuscular control that occur during PHV [15,16,17].

Therefore, the aims of the present study were: 1) to quantify the lower-limb asymmetry profile in youth tennis players through different tests (e.g., jump, dynamic balance, and CODS) and, 2) to differentiate the level of asymmetry between maturational stages and chronological groups in youth tennis athletes. It was hypothesized that youth tennis players would show increased asymmetry when grouped by maturational stages compared to chronological groups, with the largest inter-limb differences seen circa-PHV.

## 2. Material and Methods

### 2.1. Study Design

The present study used a battery of fitness tests: single leg countermovement jump (SLCMJ), star excursion balance test (SEBT), and CODS to determine the inter-limb asymmetries of youth tennis athletes. The sample was divided according to their chronological age in U18, U16, U14, U12 and maturational stages in pre-PHV, circa-PHV, and post-PHV, enabling a comparison to be drawn across age groups for physical performance and between-limb differences. All tests were conducted on the same day and in a randomized order, with players required to attend a familiarization session beforehand to understand the test protocols. During the familiarization session, athletes were allowed to practice the tests an unlimited number of times, until a satisfactory level of technical competence had been reached, while being monitored throughout. All athletes were asked to continue with their normal dietary and sleeping habits and refrain from any strenuous physical activity for 24-h prior to the test day.

### 2.2. Participants

Forty-one elite tennis players volunteered to participate in the present study, with their anthropometric data re shown in Table 1. Participants were excluded if they had incurred an injury over the three months before testing procedures. Written informed consent and assent were signed by participants and their parents or guardians (for those under 18). All volunteers were informed of possible risks and benefits related to the intervention process. The study was approved by the Catalan Sports Council Ethics Committee (07/2017/CEICGC). The experimental protocol was in accordance with the Declaration of Helsinki for the study of human subjects.

### 2.3. Procedures

Prior to the experimental session, all players performed a standardized 10-min warm up. This consisted of dynamic stretches, such as multi-planar lunges, inchworms, bodyweight squats, and spidermans and practice trials for all test protocols. Each test was practiced three times at 60, 80, and 100% of each athlete’s perceived maximum effort. A three-minute rest period was prescribed between the warm up and the first test. Jump and balance testing were performed indoors, and the CODS testing was performed on an outdoor tennis court (hard surface).

The sample was divided according to their chronological age in U18, U16, U14, U12 and maturational stages in pre-PHV, circa-PHV, and post-PHV. To calculate the biological age, four variables were registered for every subject: chronological age, stature, sitting height, and body mass. These variables were used to calculate the PHV following the formula proposed by Mirwald et al. [18]. Early maturing (pre-PHV), defined as preceding the average age of PHV by 1 year; average maturing (circa-PHV), ±1 year from PHV; and late maturing, greater than 1 year after PHV (post-PHV).

#### 2.3.1. One Hundred and Eighty Degree Change of Direction Speed (CODS) Test

Subjects were instructed to perform a single 180° CODS, for a total distance of 10 m, with the CODS occurring after a distance of 5 m (Figure 1A). Total time during the CODS test was measured with infrared beams from photocells, placed on the starting line and connected to a computer (Chronojump Boscosystem, Barcelona, Spain). The fastest time of the three trials for each leg was used for data analysis. Each trial was separated by a 60-s recovery period. A trial was considered successful if the entire foot passed the line while changing direction.

#### 2.3.2. Single Leg Countermovement Jump (SLCMJ)

Subjects were instructed to stand on one leg with the hands on hips, and the alternating leg flexed to approximately 90° at the hip and knee. Upon instruction, subjects were instructed to perform a countermovement to a self-selected depth before accelerating as fast as possible into a vertical jump (Figure 1B). The trial was disregarded and repeated if the subject was helped by the impulse of the opposite leg, did not keep the jumping leg fully extended during the flight phase of the jump or if hands came off the hips. Three successful trials per limb were collected in a randomized order. Each trial was separated by a 60-s recovery period. The height in centimeters was calculated by a contact mat system (Chronojump Boscosystem, Barcelona, Spain). The highest jump for each leg was recorded and used for subsequent data analysis.

#### 2.3.3. Star Excursion Balance Test (SEBT)

The SEBT was performed with socks on the feet as described by Overmoyer et al. [19] in three specific directions (anterior [A], posteromedial [PM], and posterolateral [PL]) (Figure 1C). The distal aspect of the subject’s big toe was centered at the junction of the grid. While maintaining a single-leg stance, the subject was asked to reach lightly with the contralateral leg in the three specific directions with hands placed on hips at all times. The maximal reach distance was measured at the point where the most distal part of the foot touched the line. The trial was disregarded and repeated if the subject failed to maintain a unilateral stance, lifted or moved the stance foot from the grid, touched down with the reach foot, or failed to return the reach foot to the starting position. The greatest reach (in centimeters) of three for each direction was used for data analysis. With these three scores, the total value of the test was calculated for these authors, using the following formula taking into account the length (cm) of the leg (LL): {[(A + PM + PL)/(LL × 3)] × 100} = SEBT composite [19].

### 2.4. Statistical Analysis

The Kolmogorov–Smirnov test was applied to determine whether data sets were normally distributed. The absolute values of the tests were normally distributed, whereas inter-limb asymmetry values were not; thus, they were log-transformed before subsequent statistical analysis. Paired samples *t*-tests were used to detect performance differences between the strongest and weakest limbs for the whole sample, introducing each test performance absolute values as dependent variables. Cohen’s *d* effect sizes were performed on pairwise comparisons which were computed as the mean difference divided by the pooled standard deviation and interpreted as: small (<0.2), moderate (0.2–0.5), and large (>0.8) mean differences [20]. The lower and upper limits for 95% confidence intervals (CI) were presented for the sample mean difference of each task when a pairwise or main effect comparison was detected.

Analysis of covariance (ANCOVA) was used to examine PHV between-group asymmetries for all neuromuscular capacities assessed, controlling for gender (the inter-limb asymmetry index for each test was introduced as the dependent variables, whereas the PHV groups were included as the independent variable). Partial Eta squared (η^2^p) scores were calculated as measures of size effect for all ANCOVA effects, where <0.01 = trivial; 0.01 to 0.06 = small; 0.06 to 0.14 = medium, and >0.14 = large [21]. The level of significance was set at *p* < 0.05. All the analyses were performed using IBM SPSS Statistics for Windows version 22.0 (IBM Corp, Armonk, NY, USA). Finally, to calculate inter-limb asymmetry between legs, the following formula was used: [(strongest−weakest/strongest) × 100] [22].

## 3. Results

Performance differences in each test between limbs for the whole sample are presented in Table 2. Differences between strongest and weakest limbs were found across all tasks (*p* < 0.001). Mean values of asymmetry showed that the SLCMJ test produced the largest between-limb differences (14.7%), whereas the CODS test showed the lowest percentage of asymmetry (2.1%). Inter-limb differences in all other tests ranged between 3.5 and 5.5%.

Mean values of asymmetry for each functional performance test according to chronological age are shown in Table 3. The between-group analysis showed differences with large age effects for the asymmetry scores of SEBT-PL (F = 3.35(3,36); *p* = 0.03; η^2^p = 0.22) and SEBT composite (F = 4.01(3,36); *p* < 0.01; η^2^p = 0.25). Post-hoc analyses showed that SEBT-PL and SEBT composite asymmetries decreased with age since U12 group had higher mean asymmetry values compared to the U16 group (respectively: 9.67 ± 3.79% vs. 4.07 ± 3.66%, CI = 0.85–2.44, *p* = 0.03 and 6.09 ± 2.14% vs. 2.13 ± 1.49%, CI = 0.19–1.85, *p* < 0.01). CODS, SEBT-PL, SLCMJ, SEBT-A, and SEBT-PL did not show differences statistically between-groups (all *p* > 0.05). The biggest asymmetries found was related to jumping action (SLCMJ) in all age groups (12–17%).

Mean values of asymmetry for each functional performance test according to maturational status are shown in Table 4. The between-group analysis showed differences with large maturational stage effects for the asymmetry scores of SEBT-PM (F = 4.53(2,37); *p* = 0.02; η^2^p = 0.20) and SEBT composite (F = 6.02(2,37); *p* < 0.01; η^2^p = 0.24). Post-hoc analyses showed that SEBT-PM and SEBT composite asymmetries decreased with increasing maturational stage. Pre-PHV group had higher mean values of asymmetry in SEBT-PM in comparison to post-PHV (6.94 ± 3.53% vs. 2.87 ± 3.12%, CI = 0.94–7.19, *p* = 0.01). In addition, the pre-PHV group had higher mean values of asymmetry in SEBT composite in comparison to the post-PHV and circa PHV groups (6.08 ± 1.82% vs. 2.97 ± 1.49% CI = 1.51–4.70; *p* = 0.001 and 2.73 ± 2.57% CI = 1.33–5.36, *p* = 0.003). CODS, SLCMJ, SEBT-A, and SEBT-PL did not show statistically differences between-groups (all *p* ≥ 0.05), but SEBT-PL was at the limit for significance (*p* = 0.05). The highest asymmetry found was related to jumping action (SLCMJ) in all maturational stage groups (12–19%), with the larger value in circa PHV group (19.31 ± 12.19).

## 4. Discussion

The present study aimed to quantify inter-limb asymmetries in jump, balance, and CODS tests and to differentiate the level of asymmetry between chronological and maturational groups in elite youth tennis players. Differences were found among groups only in the SEBT, both in maturational and chronological groups, indicating a lack of results to confirm our previous hypothesis. As such, maturational stages cannot differentiate larger levels of asymmetry among youth tennis players when comparing to a chronological division of this population. Despite these results, there are some important aspects requiring further discussion.

Larger asymmetries in SEBT composite were found in pre-PHV and U12 groups (~6%). These values were considerably higher than those found in adults [19] and youth female basketball players post-PHV [23]. Both chronological and maturational analysis showed a marked trend to detect larger asymmetries before and during PHV (U12 and U14 groups when considered chronologically). This tendency of balance asymmetry to decrease with maturation is similar to those previously reported in youth soccer players [24]. The SEBT assesses balance through the demand of forced positions, which also requires high levels of strength in the supporting limb to allow better scores [25]. Considering these results, SEBT can be considered a useful assessment tool to detect inter-limb asymmetries. Despite this, it should be acknowledged that this test is not entirely representative of the sporting actions which occur in tennis.

Greater asymmetries were found in the SLCMJ compared to the SEBT and CODS tests, and were also found when differentiating between groups both ways (i.e., maturational and chronological). This is in agreement with recent studies in other sporting populations [8,25,26]. The greater asymmetries of this test were found in circa-PHV (~19%), similar to the maturation stages comparison found in elite youth soccer [24]. Although the SLCMJ test appears to be the most sensitive test at detecting asymmetries, and is in agreement with previous research [8,25,26], it is necessary to have a critical perspective if we want to consider this value as a risk factor for injury or reduced sporting performance. Given the prevalence of lateral movements in tennis [4], tests which aim to determine functional deficits in performance in this plane of motion must also be considered, such as CODS tests.

In contrast, the CODS test revealed the lowest level of asymmetry, also in agreement with previous studies of 1.21% in female youth basketball players [23] and 2.74% in male team sport athletes [27]. In the present study, it is important to point out the biggest magnitude of asymmetry in this test was when maturational analysis was applied, specifically in the pre-PHV group. However, this value was still very small (3.18 ± 1.91), highlighting that total time may be a poor metric when looking to detect existing side-to-side differences. When considering chronological age, the U12 and U14 groups showed even lower values (2.69 ± 2.36 and 2.52 ± 2.24, respectively) than the pre-PHV and PHV groups (3.18 ± 1.91 and 2.83 ± 3.02, respectively); thus, all groups showed near perfect symmetry when using total time as a means of quantifying asymmetry. Consequently, practitioners should consider alternative test methods when looking to detect inter-limb asymmetry during CODS actions. For example, strategy-based metrics associated with CODS performance could be considered which aim to isolate the change of direction actions themselves [28]. Furthermore, during maturation, youth athletes can enhance power and speed qualities; however, this is frequently accompanied by a decrease in neuromuscular control [29]. This evidence supports the importance of neuromuscular control training [30] to reduce asymmetry, which has been suggested in previous studies [8,24].

This study was not without some limitations. First, the pre-PHV and U12 groups had a low number of subjects (*n* = 8 and *n* = 6, respectively), which in turn may have impacted the statistical power of some of the analyses. Comparing inter-limb asymmetry scores across multiple tests and chronological and maturational groups is scarce; thus, the present study helps to build on a relatively unexplored area of the literature. However, future research should aim to establish larger group sizes where possible, which may provide normative data for asymmetry at different stages of maturation. Second, although useful for the sport of tennis, these results can only be attributed to this sport. Recent research has highlighted the individual nature of asymmetries [31]; thus, practitioners are encouraged to calculate existing imbalances in their own population of athletes to determine its relevance to performance and potential injury risk.

## 5. Conclusions

Taking into consideration our results, we can conclude that chronological and maturational analysis of inter-limb asymmetries did not show different results when studying inter-limb asymmetries. When analyzing specific tests, we can detect interesting aspects when maturational stages are studied in comparison with chronological groups. Second, the SLCMJ showed the greatest magnitude of asymmetry and can be considered a useful test when aiming to detect between-limb imbalances. Third, it is necessary to develop more accurate analyses of CODS tests, given that total time appears to be a poor metric for detecting asymmetry. Finally, the largest values of neuromuscular asymmetries were shown in pre-PHV/U12 and in circa-PHV/U14, highlighting SLCMJ in circa-PHV/U14.

## Figures and Tables

**Figure 1 sports-07-00106-f001:**
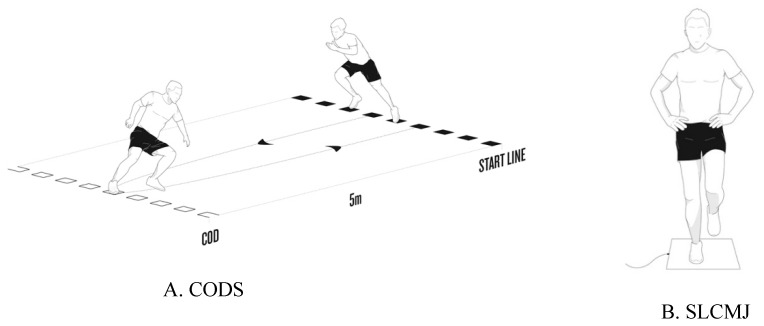
Functional neuromuscular test. (**A**). change of direction speed 180° (CODS); (**B**). single leg countermovement jump (SLCMJ); (**C1**). star excursion balance test anterior (SEBT-A); (**C2**). star excursion balance test posterolateral (SEBT-PL); (**C3**). star excursion balance test posteromedial (SEBT-PM).

**Table 1 sports-07-00106-t001:** Descriptive characteristics of the participants (*n* = 41).

	Total (*n* = 41)	Post-PHV (*n* = 20)	Circa PHV (*n* = 13)	Pre-PHV (*n* = 8)
Chronological age (years)	14.6 ± 2.7	16.4 ± 1.5	14.2 ± 1.3	10.5 ± 1.9
Peak height velocity (PHV) *	0.8 ± 2.3	2.6 ± 1.1	0.2 ± 0.4	−2.9 ± 1.2
Training experience (years)	6.6 ± 3.3	8.6 ± 2.9	5.2 ± 2.7	3.7 ± 1.6
Height (m)	1.67 ± 0.10	1.75 ± 0.10	1.67 ± 0.80	1.46 ± 0.10
Body mass (kg)	56.3 ± 13.3	65.0 ± 10.1	54.2 ± 6.3	38.1 ± 8.5
BMI (kg/m^2^)	20.0 ± 2.1	21.2 ± 1.9	19.5 ± 1.2	17.7 ± 1.8
Height seated (cm)	83.3 ± 7.0	87.7 ± 5.0	82.7 ± 3.3	73.1 ± 4.4
Length leg (cm)	91.6 ± 8.1	95.0 ± 6.1	93.0 ± 4.9	80.5 ± 7.2

Data shown as mean ± SD. * Estimation of biological age [18].

**Table 2 sports-07-00106-t002:** Strongest and weakest limbs performance of each test for the whole sample of tennis players (*n* = 41).

	Strongest	Weakest	Mean Difference (95% CI)	Asymmetry (%)	*p* Value	ES
SLCMJ (cm)	14.55 ± 4.87	12.33 ± 4.46	2.05 (1.58 to 2.51)	14.71 ± 10.05	<0.001	0.48
SEBT-A (cm)	76.99 ± 7.51	73.31 ± 7.48	3.68 (2.90 to 4.45)	4.76 ± 3.16	<0.001	0.49
SEBT-PM (cm)	114.21 ± 9.14	109.37 ± 9.51	4.84 (3.56 to 6.10)	4.22 ± 3.54	<0.001	0.52
SEBT-PL (cm)	108.33 ± 8.61	102.39 ± 9.17	5.94 (4.58 to 7.29)	5.49 ± 3.95	<0.001	0.67
SEBT composite (cm)	99.16 ± 7.05	95.71 ± 7.34	3.46 (2.75 to 4.16)	3.49 ± 2.29	<0.001	0.48
CODS (sec)	2.78 ± 0.23	2.85 ± 0.24	0.08 (0.06 to 0.09)	2.09 ± 2.24	<0.001	0.30

CI: confidence intervals; ES: Cohens’ *d* effect size; SLCMJ: single leg countermovement jump; SEBT-A: star excursion balance test anterior; SEBT-PM: star excursion balance test posteromedial; SEBT-PL: star excursion balance test posterolateral; CODS: change of direction speed 180°. Performance differences in each test between limbs for the whole sample were assessed by paired *t*-test.

**Table 3 sports-07-00106-t003:** Asymmetry index between legs profile by chronological groups.

	U18 (*n* = 12)	U16 (*n* = 13)	U14 (*n* = 10)	U12 (*n* = 6)	F	*p* *	η^2^p
SLCMJ	12.34 ± 7.72	15.31 ± 11.64	17.55 ± 9.90	13.43 ± 11.94	0.71_(3,36)_	0.55	0.06
SEBT-A	4.21 ± 2.31	3.30 ± 2.34	6.01 ± 3.28	6.89 ± 4.55	2.60_(3,36)_	0.07	0.18
SEBT-PM	2.94 ± 3.45	3.82 ± 3.46	5.37 ± 3.50	5.74 ± 3.69	1.66_(3,36)_	0.19	0.12
SEBT-PL	4.53 ± 3.47	4.07 ± 3.66	5.70 ± 3.54	9.67 ± 3.79 ^	3.35_(3,36)_	0.03	0.22
SEBT composite	3.28 ± 1.56	2.13 ± 1.49	3.99 ± 2.73	6.09 ± 2.14 ^	4.01_(3,36)_	0.01	0.25
CODS	1.31 ± 1.21	2.21 ± 2.89	2.52 ± 2.24	2.69 ± 2.36	1.44_(3,36)_	0.25	0.11

* Significantly different between subjects analyzed by ANCOVA; ^ significantly different from U16 (*p* < 0.05). SLCMJ: single leg countermovement jump; SEBT-A: star excursion balance test Anterior; SEBT-PM: star excursion balance test posteromedial; SEBT-PL: star excursion balance test posterolateral. CODS: change of direction speed 180°; η^2^p: partial eta-squared.

**Table 4 sports-07-00106-t004:** Asymmetry index between legs profile by maturational stage groups.

	Post-PHV (*n* = 20)	Circa PHV (*n* = 13)	Pre-PHV (*n* = 8)	F	*p* *	η^2^p
SLCMJ	12.53 ± 7.17	19.31 ± 12.19	14.49 ± 10.93	1.27_(2,37)_	0.29	0.06
SEBT-A	4.12 ± 2.56	5.52 ± 4.19	5.27 ± 2.55	0.80_(2,37)_	0.46	0.04
SEBT-PM	2.87 ± 3.12	4.63 ± 3.32	6.94 ± 3.53 ^	4.53_(2,37)_	0.02	0.20
SEBT-PL	4.21 ± 3.29	5.79 ± 4.61	7.98 ± 3.45	3.13_(2,37)_	0.05	0.14
SEBT composite	2.97 ± 1.49	2.73 ± 2.57	6.08 ± 1.82 ^ ^#^	6.02_(2,37)_	<0.01	0.24
CODS	1.12 ± 1.30	2.83 ± 3.02	3.18 ± 1.91	1.96_(2,37)_	0.16	0.09

* Significantly different between subjects analyzed by ANCOVA; ^ significantly different from Post-PHV (*p* < 0.05) and ^#^ significantly different from PHV after Bonferroni adjustment (*p* < 0.01). SLCMJ: single leg countermovement jump; SEBT-A: star excursion balance test anterior; SEBT-PM: star excursion balance test posteromedial; SEBT-PL: star excursion balance test posterolateral. CODS: change of direction speed 180°; η^2^p: partial eta-squared.

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
