# Peer review of "Effects of Maturation on Lower Limb Neuromuscular Asymmetries in Elite Youth Tennis Players"

_sports, 2019, doi:10.3390/sports7050106_

Round 1
Reviewer 1 Report
This study examined maturation effects in tennis players. I have some issues that need to be addressed by the authors:
1. Dominance was measured as the stronger side and this is not entirely correct.
2. The Discussion and conclusions are in contrast to the author’s expectations and it is clear that they wrote the Discussion as there were asymmetries. Not finding group differences in asymmetries is equally important as finding differences. Consequently, the Discussion and Conclusion should be completed revised accordingly.
Specific comments
Line 38: Define “asymmetric” sport
Line 39: “4” is a reference or a number?
Line 39: The first time you refer to neuromuscular assymetries you need to provide which variables you are refer to i.e. strength, neural activation ? assymetries between limbs? Between muscles?
Line 43: “greater lower limb-assymetries” symmetries in what? Further, do you mean between limbs?
Line 45: change “youth” into “young”
Line 46: add years in age
Line 46: “lateral hop test” do you mean score?
Line 51: Define “COD”
Line 106: This is a rather arbitrary definition of dominance. Please provide a rationale for using this method to define dominance, Dominance is a more complex concept and it is not certain that the stronger limb is the dominant one.
Lines 136-143: This is a very demanding task, especially for young children. Was there any familiarization or training provided prior to the main testing session?
Figure 1.C. is not visible, at least in my pdf version.
Line 156: Please re-state in the statistics section, that the dependent variable was inter-limb asymmetric index or if absolute values were compared.
Line 158: If data were normally distributed please state so. If some are not normally distributed, please state so.
Table 2: I cannot understand what these comparisons stand for. Do you refer to differences in performance in each tests between limbs for the whole sample? If yes, please re-organize the statistics and results sections (perhaps by inserting a paragraph), so that the reader can follow the results.
Line 222-230: Finding no assymetries is equally important as finding assymetries. The authors should focus on their main finding, that there are minimal differences between maturation groups in assymetries and that asymmetry indices are low. Assymetries in the range of 0-5%, cannot be considered as significantly clinical assysmetries as they lie in the area of variability and expected inter-limb difference for various reasons. We cannot be absolutely symmetrical, some asymmetry is expected and it is not abnormal.
Line 258: The results show no maturation differences. Consequently, this conclusion is not correct. Lines 259-265 try to explain a hypothesis that is not supported by your results. Please revise.
Conclusion: The conclusion should be shortened and it should focus on the results of the study. Your study showed minimal maturation differences, therefore, your conclusion section cannot recommend that maturation should be an issue for future studies. The last 3-5 sentences are not conclusions but discussion and should be omitted.
Author Response
Response to Reviewer 1 Comments
This study examined maturation effects in tennis players. I have some issues that need to be addressed by the authors:
1. Dominance was measured as the stronger side and this is not entirely correct.
2. The Discussion and conclusions are in contrast to the author’s expectations and it is clear that they wrote the Discussion as there were asymmetries. Not finding group differences in asymmetries is equally important as finding differences. Consequently, the Discussion and Conclusion should be completed revised accordingly.
Thanks for the comments. We respond to specific comments
Specific comments
Line 38: Define “asymmetric” sport
Thank you for your comment. We define ‘asymmetric sport. In relation to repetitive asymmetrical actions in tennis. Please see lines 41
Line 39: “4” is a reference or a number?
Sorry, it’s a mistake. We provide the changes. Please see lines 42
Line 39: The first time you refer to neuromuscular assymetries you need to provide which variables you are refer to i.e. strength, neural activation ? assymetries between limbs? Between muscles?
Thank you for your comment. Many authors have defined neuromuscular control (Heweet et al. 2003; Lephart et Rienman et al., 2002), in relation to precise muscle activation to perform a coordinated movement. We consider that literature is extensive at this point (Bishop et al. 2017, 2018; Fort-Vanmerhaegee et al. 2015)
Line 43: “greater lower limb-assymetries” symmetries in what? Further, do you mean between limbs?
Thank you for your comment.. We provide the changes. Please see line 48
Line 45: change “youth” into “young”
Thank you for your comment.. We provide the changes. Please see line 49
Line 46: add years in age
Thank you for your comment.. We provide the changes. Please see line. 50-53
Line 46: “lateral hop test” do you mean score?
Thank you for your comment.. We provide the changes. Please see line 51-52
Line 51: Define “COD”
Thank you for your comment.. We provide the changes. Please see line 57
Line 106: This is a rather arbitrary definition of dominance. Please provide a rationale for using this method to define dominance, Dominance is a more complex concept and it is not certain that the stronger limb is the dominant one.
Thank you for your comment. We can understand this point of view if we consider some aspects related to neuromuscular control and neuromuscular coordination of movements, but literature consistently identify the stronger limb as the dominant one (Bishop et al. 2018, Dos’Santos et al. 2017; ) thinking on performance.
Lines 136-143: This is a very demanding task, especially for young children. Was there any familiarization or training provided prior to the main testing session?
Thank you for your comment. We can provide this explanation in study design section
Figure 1.C. is not visible, at least in my pdf version.
We will review the PDF, since in the original version the drawing of the SEBT test is provided
Line 156: Please re-state in the statistics section, that the dependent variable was inter-limb asymmetric index or if absolute values were compared.
Thank you for the suggestion. It has been detailed. Please see lines 177 and 180, page 6
Line 158: If data were normally distributed please state so. If some are not normally distributed, please state so.
The inter-limb asymmetric index of all tests performed were not normally distributed. It has been stated in the manuscript as suggested. Please see line 1638 to 172, page 6
Table 2: I cannot understand what these comparisons stand for. Do you refer to differences in performance in each tests between limbs for the whole sample? If yes, please re-organize the statistics and results sections (perhaps by inserting a paragraph), so that the reader can follow the results.
We used paired samples t-tests to detect performance differences between the dominant and non-dominant limbs of all participants as a group. It was explained in line 170 but we have rephrased it slightly in order to be clearer. We referred to table 2 in the results sections when summarizing these analyses, but we have inserted some text to make clearer in the results section that the comparisons between limbs were performed in the whole sample. We have done so in the table caption too. Thank you for the appreciation. Please see line 165 to 197, page 6 and table 2.
Line 222-230: Finding no assymetries is equally important as finding assymetries. The authors should focus on their main finding, that there are minimal differences between maturation groups in assymetries and that asymmetry indices are low. Assymetries in the range of 0-5%, cannot be considered as significantly clinical assysmetries as they lie in the area of variability and expected inter-limb difference for various reasons. We cannot be absolutely symmetrical, some asymmetry is expected and it is not abnormal.
Thank you for your comment, but, as we write at the beginning of the discussion section, as well as the treatment we do to the conclusions, we expos the general lack of differences. According to our results and the way we explain them, we consider we express the idea of the reviewer.
Line 258: The results show no maturation differences. Consequently, this conclusion is not correct. Lines 259-265 try to explain a hypothesis that is not supported by your results. Please revise.
Thank you for your comment, but, We would like to explain that the main asymmetries found in U12 and pre-PHV may be a limitation of sporting abilities (due to the effects of neuromuscular asymmetries in performance, explained above) and lower neuromuscular control (defined in the document) in these ages.
In order to clarify and be more cautious, we changed this affirmation.Please see lines 271
Conclusion: The conclusion should be shortened and it should focus on the results of the study. Your study showed minimal maturation differences, therefore, your conclusion section cannot recommend that maturation should be an issue for future studies. The last 3-5 sentences are not conclusions but discussion and should be omitted.
Thank you very much for your comments, we have deleted the last 4 sentences.We have provided the stage of the chronological age, to highlight the variations of asymmetries in relation to chronological or biological age. Please see lines 102
Reviewer 2 Report
This paper is a very important research paper from the preventive medicine point of view on research on growth and laterality. However, please explain the following points.
Why is there a negative value of PHV in column of Pre-PHV group in Table 1? What is the unit of PHV?
Author Response
Response to Reviewer 2 Comments
This paper is a very important research paper from the preventive medicine point of view on research on growth and laterality. However, please explain the following points.
Why is there a negative value of PHV in column of Pre-PHV group in Table 1? What is the unit of PHV?
Thanks for your comments. As Mirwald et al. 2002 showed any negative maturity offset prediction should classify the individual as pre-PHV and any positive prediction as post-PHV. Negative value in pre-PHV relates to the time it from a PHV stage
Round 2
Reviewer 1 Report
The authors have made an attempt to address the comments. In my opinion, however, these revisions were unsatisfactory. Namely,
1. The issue of dominace has not been addressed adequately. The fact that a previous study has used the "strongest" leg as the "dominant" is not necessarily correct. Hence, the authors could have done a better job, to explain their decision and revision accordingly, They could replace "dominant" with "stronger" or state this in a limitation.
2. The authors did not find significant assymetries, as they clearly state in the first paragraph of the Discussion. Nevertheless, they then discuss trends and potential assymetries and make general conclusions, highlighting assymetries. Although this was highligted in my previous comments, there Discussion was not revised, as requested.
3. Additionally, the term "neuromuscular" assymetries suggests that measures of neuromuscular activation such as EMG were implemented which is not entirely correct. The authors are adviced to replace "neuromuscular performance tests" with "field-tests" or "functional performance tests" or something similar, as previous studies have done.
Specific comments
;Line 41: This is not a definition of assymetry. It is the cause of assymetry and it is rather speculative.
Line 48: There is no such thing as jump assessments. It is vertical jump height or power or a specific variable.
Line 306-307: This is in conrtrast to the first paragraph of the Discussion. The whole practical applications section should be revised based on the results.
Author Response
The authors have made an attempt to address the comments. In my opinion, however, these revisions were unsatisfactory. Namely,
1. The issue of dominace has not been addressed adequately. The fact that a previous study has used the "strongest" leg as the "dominant" is not necessarily correct. Hence, the authors could have done a better job, to explain their decision and revision accordingly, They could replace "dominant" with "stronger" or state this in a limitation.
Thank you very much for the comments, to make the concept clearer, and to have more consistency with our intention, we have related the changes that the reviewer suggests and we will refer to the strongest and the weakest Please see line 21 and in the rest of the document
2. The authors did not find significant assymetries, as they clearly state in the first paragraph of the Discussion. Nevertheless, they then discuss trends and potential assymetries and make general conclusions, highlighting assymetries. Although this was highligted in my previous comments, there Discussion was not revised, as requested.
Dear reviewer,
Related to this issue, we have decided to rewrite the first paragraph of the discussion 233-240 and other paragraphs from the discussion 263-279. Related to conclusions please see lines 295-298
3. Additionally, the term "neuromuscular" assymetries suggests that measures of neuromuscular activation such as EMG were implemented which is not entirely correct. The authors are adviced to replace "neuromuscular performance tests" with "field-tests" or "functional performance tests" or something similar, as previous studies have done.
Response: Thanks for your comment, we have replaced the expression 'neuromuscular' when we refer to the assessments, and we have preferred use the expression 'functional performance test', which the reviewer has suggested Please see lines 203
Specific comments
Line 41: This is not a definition of assymetry. It is the cause of assymetry and it is rather speculative.
Thank you very much for your comment, to make our intention clearer, we have removed the term asymmetric sport, and we have highlighted the greatest number of player actions on one side in relation to another, and its effects (according to bibliography) Please see lines 42
Line 48: There is no such thing as jump assessments. It is vertical jump height or power or a specific variable.
Response: Thank you for your comment, we made the changes. Please see lines 49
Line 306-307: This is in contrast to the first paragraph of the Discussion. The whole practical applications section should be revised based on the results.
Dear reviewer, We decide remove this section.
The English language and style was checked